# Peering inside the black box by learning the relevance of many-body functions in neural network potentials

Klara Bonneau [1,9], Jonas Lederer [2,3,9], Clark Templeton [1,9] [envelope], David Rosenberger [1,4], Lorenzo Giambagli [1], Klaus-Robert Müller [2,3,5,6] [envelope] & Cecilia Clementi [1,7,8] [envelope]

Machine learned potentials based on artificial neural networks are becoming a popular tool to define an effective energy model for complex systems, either incorporating electronic structure effects at the atomistic resolution, or effectively renormalizing part of the atomistic degrees of freedom at a coarse-grained resolution. One main criticism regarding neural network potentials is that their inferred energy is less interpretable than in traditional approaches, which use simpler and more transparent functional forms. Here we address this problem by extending tools recently proposed in the nascent field of explainable artificial intelligence to coarse-grained potentials based on graph neural networks. With these tools, neural network potentials can be practically decomposed into n-body interactions, providing a human understandable interpretation without compromising predictive power. We demonstrate the approach on three different coarse-grained systems including two fluids (methane and water) and the protein NTL9. The obtained interpretations suggest that well-trained neural network potentials learn physical interactions, which are consistent with fundamental principles.

Molecular simulations have emerged in the last 75 years as a valuable tool to recover or even predict interesting physical phenomena at the microscopic scale and provide a detailed mechanism for grasping the underlying molecular processes[1]. In principle, the most accurate description of a molecular system is given by the solution of the associated Schrödinger's equation. However, it is common practice to invoke the separation of scales between electrons and nuclei (Born-Oppenheimer approximation) and define an effective energy function for the nuclei that should take into account the electronic effects[2]. Historically, this has been done empirically in the definition of classical atomistic force-fields, which have been designed, refined, and used for the study of molecular systems[3,4]. Classical force-fields assume that the

potential energy of a molecular system can be described as a function of "bonded" terms (e.g. bonds, angles, dihedrals) and "non-bonded" pairwise potentials (e.g., Van der Waals, Coulomb)[1,2]. All the potential energy terms are defined by fixed functional forms, with parameters tuned to reproduce experimental data and/or first principle calculations on small test systems[1,3].

Recent advances in machine learning (ML) have triggered a step-change in the development of data-driven force-fields. In particular, neural network potentials (NNPs) have been proposed to more accurately capture the electronic effects in the potential energy functions for the nuclei[5,6]. While classical, non-bonded potential terms are generally limited to 2-body interactions, the use of NNPs, and more

[1]Department of Physics, Freie Universität Berlin, Berlin, Germany. [2]Machine Learning Group, Technische Universität Berlin, Berlin, Germany. [3]BIFOLD - Berlin Institute for the Foundations of Learning and Data, Berlin, Germany. [4]BAM Federal Institute for Materials Research and Testing, Berlin, Germany. [5]Department of Artificial Intelligence, Korea University, Seoul, South Korea. [6]Max Planck Institute for Informatics, Saarbrücken, Germany. [7]Center for Theoretical Biological Physics, Rice University, Houston, TX, USA. [8]Department of Chemistry, Rice University, Houston, TX, USA. [9]These authors contributed equally: Klara Bonneau, Jonas Lederer, Clark Templeton. [envelope]e-mail: clarktemple03@gmail.com; klaus-robert.mueller@tu-berlin.de; cecilia.clementi@fu-berlin.de

specifically graph neural networks (GNNs)[7] significantly increases the expressivity of the energy function and allows flexible parameterization of many-body interactions[8–13].

While leaps in the development of GNN-based models have shown great promise in studying complex macromolecular systems[14] and predicting material properties[15], the results and the models themselves are often seen as black boxes. NNPs take molecular conformations as input and only provide the corresponding potential energy and its derivatives as output. The increased model accuracy comes at the cost of insight into the nature and strength of molecular interactions. In a classical force-field each term in the energy function can be dissected, but deciphering which terms in the potential energy are important for stabilizing certain physical states or interpreting a prediction is significantly more difficult in a GNN-based model.

In parallel to the development of atomistic NNPs, GNNs have been successfully employed in the definition of models at reduced resolutions[16–21], where some of the atomistic degrees of freedom are renormalized into a reduced number of effective "beads" to speed up the simulation time. The difficulty in the definition of coarse-grained (CG) models lies in the fact that many-body terms play an important role, as a reduction in the number of degrees of freedom is associated with increased complexity in the effective CG energy function. It has been shown that, to reproduce experimentally measured free energy differences[22] or the thermodynamics of a finer-grained model[23,24], many-body terms need to be included. Like atomistic NNPs, CG NNP models exhibit a black-box nature, offering no insight into the learned many-body terms. Given the necessity and complexity of capturing these terms, the development of interpretable CG NNPs is highly desirable.

The black-box problem is not unique to molecular systems, but rather ubiquitous in the application of ML. For trusting image classifiers, it is important to know if an accurate classification stems from a correct learning of the features or a learning of an undetected bias in the training set (e.g. refs. [25–27]). Analogously, to trust NNPs and their ability to extrapolate to new systems, it is important to know if an accurate prediction arises from the network learning the physical properties of the different interactions or is merely a data memorization, or a compensation of errors[28]. As a response, the new area of "Explainable Artificial Intelligence (XAI)" has emerged to begin providing tools to tackle the interpretation of artificial neural networks[29]. The field of XAI ranges from self-explainable architectures[11,30–32] to post-hoc explanations[33–36]. Some of those approaches are starting to find use also in physical and chemical applications, e.g., for explaining predictions regarding toxicity or mutagenicity[37–39], predictions of electronic-structure properties[40,41], guiding strategies in drug discovery[42,43], analyzing protein-ligand binding[44–46], or uncertainty attribution[47,48]. XAI approaches have also been utilized to provide a better understanding of the error introduced in molecular models[49].

In this work, we apply XAI to explain NNPs in the context of molecular dynamics simulations of CG systems. The NNPs are trained at CG resolution from atomistic simulation data for several systems of different complexity. We use Layer-wise Relevance Propagation applied to GNNs (GNN-LRP) to interpret the CG models beyond a mere energy prediction. In particular, we study the *n*-body contributions associated with the learned effective interactions in the NNP, and evaluate them based on fundamental principles. A key advantage of using GNN-LRP for this task is its ability to reveal learned interactions among a subset of beads taking into account their surroundings. On the other hand, the many-body decomposition of the model output with the traditional recursive method (e.g. ref.[28]) determines the energy needed to form isolated *n*-mers of beads from sub-elements, thus ignoring the effect of the surrounding environment.

Ideally, an "interpretable model" should enable researchers to build trust in its predictions and, in a second step, extract scientific knowledge from successful applications while identifying the sources of deficiencies or anomalies when the model fails. Here, we focus on the first step – enhancing trust in NNPs for CG systems by demonstrating the physically sound nature of the learned interactions in two examples. As a first example, we compare different classes of GNN architectures to obtain CG models of bulk fluids and show that an interpretation of accurate CG models provides meaningful insight into the learned physical concepts. Interestingly, two GNN architectures, even if quite different from each other, convey the same physical interpretation: At least in terms of 2-body and 3-body contributions, the two networks appear to offer different functional representations of the same underlying energy landscape. As a second example, we examine a machine learned CG model for the protein NTL9 and show that its interpretation allows us to pinpoint the stabilizing and destabilizing interactions in the various metastable states, and even interpret the effects of mutations. The fact that the learned interactions align with existing physical and chemical knowledge makes the employed GNNs more trustworthy, supporting a wider use of these methods.

## Results

### Application of Layer-wise Relevance Propagation to Neural Network Potentials

A computationally efficient and minimally invasive approach to explain the predictions of neural networks in a post-hoc manner is layer-wise relevance propagation (LRP)[29,34]. The purpose of LRP is to decompose the activation value of each neuron into a weighted sum of contributions from its inputs. In the case of the final layer of NNPs, this amounts to decomposing the energy output into a sum of energy contributions of its inputs. Once this decomposition is complete, the numerical values of each element in the sum can be examined, with larger absolute values considered more relevant contributions (stabilizing or destabilizing, depending on their sign) and smaller values deemed less relevant.

To fix the ideas, let us first consider a multi-variable function $f$ such that $f : \mathbb{R}^N \to \mathbb{R}$ and assume that the following decomposition is done:

$$f(\mathbf{x}) = \sum_{\alpha=1}^{N} f_\alpha(x_\alpha) \quad \text{such that} \quad |f_\alpha(x_\alpha)| \geq |f_{\alpha+1}(x_{\alpha+1})| \tag{1}$$
$$\forall \alpha \in \{1, \dots, N-1\}.$$

Then $f_1(x_1)$ and $f_N(x_N)$ are the most and least relevant elements, respectively. We could therefore state that the contribution due to the variable $x_1$ to the function is larger than the contribution due to the variable $x_N$. As each neuron in each layer of a modern neural network can be seen as a multi-variable function, such relevance decomposition can be iteratively applied (while maintaining layer-wise relevance conservation[34,50]), from the last computation to the first one, allowing for an explanation of the neural network prediction.

A straightforward approach to obtain an expression like Equation (1) is the first-order Taylor expansion around a given point $\mathbf{x}^*$. As discussed in the Methods section, different choices of the expansion point $\mathbf{x}^*$, which are deeply connected with the neural network architecture and behavior, imply different relevance propagation rules[29]. Crucially, $\mathbf{x}^*$ is not chosen to be extremely close to the input point $\mathbf{x}$, and that is why, despite being linked to Taylor decomposition and therefore explicitly depending on the derivative of the function, LRP is not a sensitivity analysis[25]. Indeed, LRP provides an explanation of the model's output by decomposing it into contributions from sub-sets of input features, whereas sensitivity analysis focuses on how local changes in input features affect the output.

For the case of GNNs, the LRP technique can be extended to eventually attributing relevance to higher-order features in the input graph. This approach is referred to as GNN-LRP[41], schematically

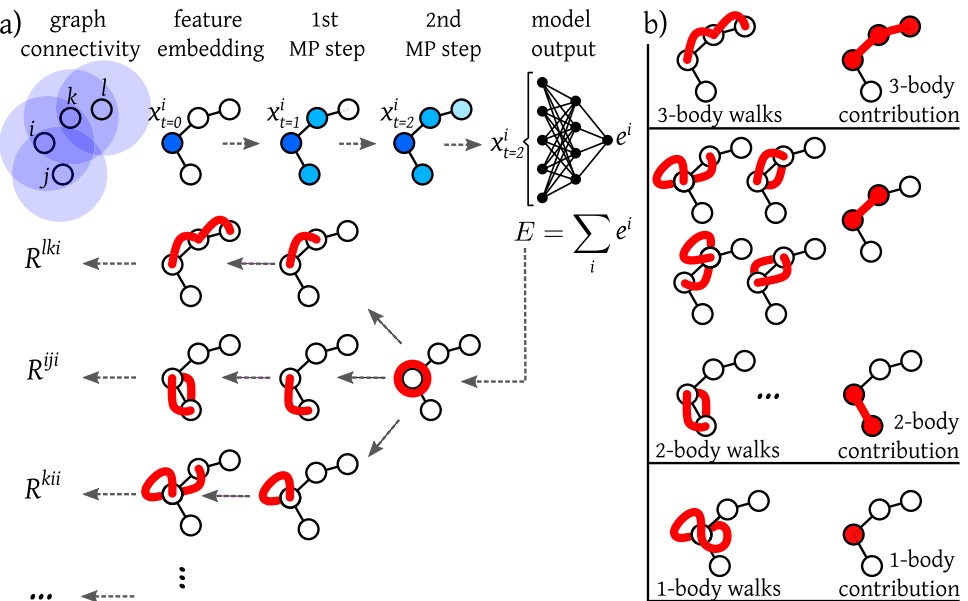

**Fig. 1 | Concept of Layer-wise Relevance Propagations for Graph Neural Networks (GNN-LRP) illustrated for a system of four particles (i.e. coarse-grained (CG) beads, in the present context). a** In Graph Neural Networks, the input graph is defined by a cutoff radius (illustrated in shaded blue) that determines the direct neighbors for each input node. By iterative message-passing (MP) steps in multiple layers (layer index $t \in \{0, 1, 2\}$), information can be exchanged between more distant nodes (outside of the cutoff region), updating the representation of each node $\mathbf{x}_t^i$. The model output (in our case the potential energy $E$) is obtained by passing the learned feature representations through a multilayer perceptron and final pooling over the individual bead energies $e^i$. Obtaining the relevance $R^{walk}$ involves propagating the output back through the network, by considering the connections between each node in one layer and the nodes in the previous layer. This procedure defines "walks" across the network layers. **b** The walks involving the same subset of $n$ nodes are aggregated to obtain a decomposition of the output into $n$-body contributions.

described in Fig. 1a. In a nutshell, GNN-LRP decomposes the model's energy output into relevance attributions to sequences of graph edges. Those sequences describe "walks" over a few nodes in the input graph. The associated relevance attribution is also often referred to as the relevance score of the walk. By aggregating the relevance scores of all walks associated with a particular subgraph, we can determine its $n$-body relevance contribution to the model output (shown in Fig. 1b). A more detailed description of GNN-LRP and of how the relevance of $n$-body contributions are calculated, is provided in the Methods Section as well as in Supplementary Section S6.

## Methane and Water

We start by analyzing and comparing CG models for bulk methane ($CH_4$) and water ($H_2O$). Methane has previously been studied with various coarse-graining methods, since its non-polar and weak Van-der-Waals interactions make it a simple test system[16]. On the other hand, water is capable of forming complex hydrogen bonding structures and much research has been devoted to its modeling, both at the atomistic scale[51] and at the CG level[17,52–56].

For both systems, we define CG models by integrating out the hydrogen atoms and positioning an effective CG bead in place of the central carbon or oxygen atom. For each system, we train two CG models with different choices of GNN architectures, PaiNN[57] and SO3Net[58], to define the CG effective energy. Both architectures are trained using the force-matching variational principle for coarse-graining, to create a thermodynamically consistent CG model from the atomistic data[18,53,59–61]. With the trained models, we perform molecular dynamics (MD) simulations for both liquids. For more details on the MD simulation behind the atomistic data or the CG simulations based on the NNPs, please refer to the Supplementary Section S1. A brief discussion of the specific features of these GNNs and more details of NNP training are provided in the Methods Section.

The ability of the two CG models to reproduce the structural features of the two systems is shown in Fig. 2 and Supplementary Figs. S1 and S7. In particular, the radial distribution function (RDF) as obtained in the CG models is shown in Fig. 2 against the atomistic reference model for methane (right column) and water (left column). Note that the SO3Net model exhibits irreducible representations up to a rotation order of $l_{max} = 2$, while PaiNN utilizes a maximum rotation order of $l_{max} = 1$. With increasing $l_{max}$, the NNP can learn more expressive representations of the chemical environment in each message-passing block and as a consequence, in comparison to SO3Net, PaiNN requires a larger cutoff to accurately reproduce the RDF of water. For more details on the difference between irreducible representations of different $l_{max}$, we refer to the Supplementary Section S3.1. For a comparison between PaiNN models with different cutoff radii, please refer to Supplementary Fig. S2.

For methane, the smooth oscillatory behavior of the RDF is similar to a Lennard-Jones (LJ) fluid[62], suggesting that many-body interactions may not be very relevant in a CG model of this molecule. In contrast, for water, the height of the first solvation shell is sharply peaked and decays more rapidly than in the case of methane. For both systems, both architectures reproduce the corresponding RDF.

In Fig. 2, the average 2-body relevance contribution is plotted (red curves) as a function of their distance, alongside the RDF for each model. Since the relevance contributions correspond to a decomposition of the output energy, a positive (negative) relevance contribution implies an increase (decrease) in the energy, thus a destabilizing (stabilizing) effect of the associated interaction.

For both methane and water, both PaiNN and SO3Net show a 2-body relevance contribution that diverges as the distance between two beads goes to zero. This observation matches our intuition that at distances below a certain "effective radius", the network should learn a repulsive excluded-volume interaction to avoid the overlapping of the CG beads. For all models, the relevance contribution decays to zero as the distance between two beads approaches the cutoff value considering two beads connected in the corresponding GNN. This corresponds to the intuition that the interactions between two beads

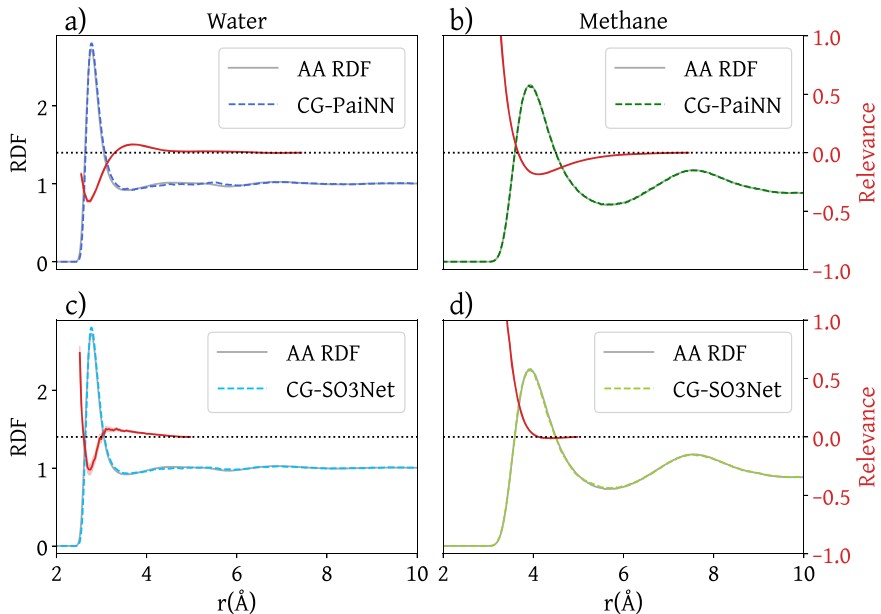

**Fig. 2 | Comparison of radial distribution functions (RDF) resulting from simulations with an atomistic (AA) or coarse-grained (CG) model and corresponding 2-body relevance.** Panels **a**, **c** correspond to water and **b**, **d** to methane models. Panels **a**, **b** show the results for PaiNN-based, and **c**, **d** for SO3Net-based models. The relevance in arbitrary units, shown in red, is normalized by the absolute total relevance over the number of walks of the respective model and rescaled for each model type. A negative value implies a stabilizing interaction as the model output is the energy. Relevance values are averaged in 75 bins across the distance range, the corresponding average is shown in a solid line and shaded regions correspond to the standard deviation in each bin.

become weaker as the beads move further apart, and is enforced by the cosine shape of the cutoff function.

The 2-body relevance contributions in all models show an inverse relationship to the RDF. In the case of water, both models display a stabilizing well near the first RDF peak corresponding to the first solvation shell. For SO3Net trained on methane, this dip in relevance is diminished due to its relatively short cutoff. For the same reason, the relevance attributions show only small or no stabilizing wells for the second solvation shells of both water and methane. Despite the effect of the cutoff function at larger distances, both models exhibit similar behavior for the two liquids.

While the 2-body contributions are very similar for all scenarios, the 3-body contributions indicate significant differences between the water and methane models. To visualize the 3-body contribution, we describe each triplet of beads by the largest angle and opposed edge length in the triangle formed by the three beads. The 3-body relevance contribution may then be visualized in a contour plot as depicted in Fig. 3. The plot shows the average 3-body contribution of triplets grouped in bins of similar angle and distance values. The 3-body contribution of each individual triplet of beads is calculated as explained in the Methods Section.

It is first interesting to note that the 3-body relevance contributions for methane are very close to zero compared to water, indicating that the 3-body terms are not very important for the CG methane. For water, both models produce similar behavior for the averaged 3-body relevance contributions. The fact that two different architectures, with different interaction cutoffs, learn a similar relevance distribution indicates that these models learn the same underlying potential of mean force.

The destabilizing contributions of the water models correspond to the shortest distances at any given angle. Here, the 3-body terms likely correct for an overstructuring of the 2-body interactions. This is corroborated by Supplementary Fig. S9, where we plot the 3-body contribution as a function of three variables entirely defining the involved triangle (largest angle and length of the two adjacent edges): For the destabilizing contributions, the length of the two smallest

edges in the triangle is comparable to the distance corresponding to the first steep increase of the RDF, indicating that these destabilizing contributions stem from the same repulsive excluded-volume interaction also learned by the 2-body terms. Having 3-body interactions correct for 2-body interactions is a known effect when parameterizing explicit n-body functions to build CG models[54,55,63]. The strongest stabilizing contributions in the water models in Fig. 3 correspond to configurations with an angle around 50-60 degrees, associated with the population of water molecules sitting interstitially inside the tetrahedral arrangement[64,65]. The fact that, in both models, 3-body terms are crucial to recover structural properties of CG water indicates that the model does not only use 3-body terms to correct for 2-body terms but also learns "real" effective 3-body terms, representing the hydrogen bonding propensity of water.

To further support the statement that the NNPs trained on water learn meaningful 3-body interactions while the NNPs trained on methane rely solely on 2-body interactions, a 2-body-only water model is shown in Supplementary Section S3, where the Inverse Monte Carlo (IMC)[66,67] method is used for parameterizing a pair potential on the system's RDF. Supplementary Fig. S1 shows that, for methane, the IMC 2-body model is still capable of reproducing the correct distributions of the relevant features, whereas for water, it fails to reproduce the angular distribution. Additionally, we also observe that an invariant but non-equivariant GNN such as SchNet[68] performs well on methane, but fails at fully reproducing the atomistic distributions for water (see Supplementary Section S3).

Supplementary Fig. S7 shows the distribution of angles and distances for triplets involved in a 3-body contribution given the specified network cutoff, both for the atomistic (training) datasets and for the CG simulations. It is interesting to see that in all four models, the NNP-CG simulations are able to reproduce the distributions from atomistic simulations. Furthermore, Supplementary Fig. S7 illustrates that, even if from the analysis of the 2-body interactions it may appear that the network was merely learning stabilizing interactions for configurations that were present frequently in the training dataset (e.g. RDF peak), the distributions of angles and distances in both atomistic and CG

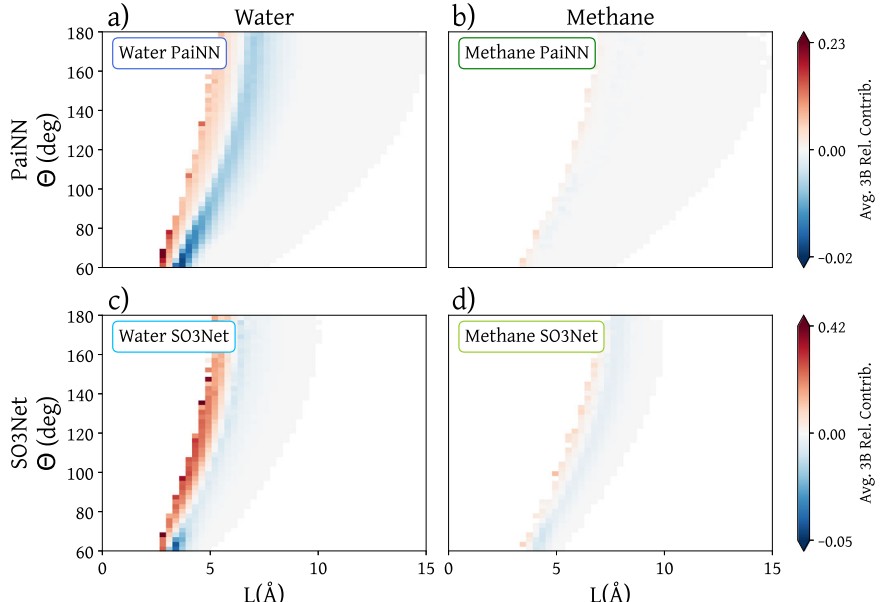

**Fig. 3 | Average 3-body relevance contributions for water and methane on PaiNN and SO3Net models as a function of largest angle ($\theta$) and opposed edge length ($L$) for triplets involved in a 3-body walk.** Panels **a**, **c** correspond to water and **b**, **d** to methane models. Panels **a**, **b** show the results for PaiNN-based, and panels **c** and **d** for SO3Net-based models. The relevance, in arbitrary units, is normalized by the absolute total relevance over the number of walks of the respective model and rescaled for each model type. A negative value implies a stabilizing interaction as the model output is the energy. Angle and distance ranges are divided in 50 bins each, and colors correspond to average values of the 3-body relevance for all triplets whose largest angle and opposed edge length correspond to the bin.

simulations show that the learned effective 3-body interactions in the NNPs represent more than simple statistics derived from the training dataset.

It is worth noting that while the average relevance contribution is shown in Figs. 2 and 3 as a function of distances and angles, the individual relevance scores for single walks over configurations of beads cover a broad range. In the Supplementary Section S5.1, we provide a more detailed analysis on the relevance attributed to individual walks. Supplementary Figs. S5 and S6 show the entire distribution of 2-body walk relevance over all the configurations of the water and methane models. High (positive, destabilizing) relevance values correspond to short distances between beads and low (negative, stabilizing) relevance values to distances located at the first peak of the RDF. A relevance score of almost zero corresponds to distances approaching the network cutoff.

The contour plots in Fig. 3 show the 3-body contributions as a function of only two variables, thus averaging out the additional variable uniquely defining the triangle formed by three interacting beads. In Supplementary Fig. S9, we show the distribution of 3-body contributions as a function of three variables entirely determining the triangle formed by the involved triplet of beads. The PaiNN model for methane shows the rare appearance of stabilizing 3-body contributions corresponding to the shortest edge of the triangle measuring 3.15 Å, on the onset region of the RDF, where the energy should be dominated by repulsive interactions. These contributions are absent in the SO3Net model and are very rare in the training distribution: Supplementary Fig. S10 shows that only very few data points correspond to these configurations, and the learned repulsive 2-body contributions are an order of magnitude stronger than these stabilizing 3-body contributions. While this model performs seemingly well in MD simulations (see the reproduction of the structural metrics in Fig. 2 and Supplementary Figs. S1 and S7), this detailed analysis of the learned contributions with GNN-LRP reveals some shortcoming of this model that could be improved by enhancing the training data distribution. More details on this analysis can be found in the Supplementary Section S5.2. This result underscores how GNN-LRP can uncover model

artifacts in the learned interactions, which cannot be seen by simply considering the thermodynamic and structural properties obtained by MD simulations.

As mentioned above, the main advantage of GNN-LRP over traditional many-body decomposition is that GNN-LRP gives $n$-body contributions of subgraphs in the environment of the entire system while the many-body decomposition framework computes the $n$-body energies, as predicted by the NNP, to form isolated $n$-mers from $m$-mers with $m < n$. As demonstrated and discussed in detail in the Supplementary Section S4, the traditional many-body decomposition of the NNP gives rise to noisy and uninformative 2- and 3-body terms as these terms are taken in isolation and not in the context of the bulk fluid, as is instead done by GNN-LRP.

Both 2-body and 3-body contributions obtained with GNN-LRP resonate with fundamental knowledge of the water and methane systems. NNPs learn 2-body interactions aligning with the RDF and learn that 3-body interactions for methane are negligible, while for water they are important. GNN-LRP especially allows us to show that, overall, models based on different architectures trained on the same data show mostly similar relevance attributions, suggesting that the different models have learned the same underlying physical interactions. Additionally, it uncovered slight model deficiencies that were not detected with MD simulations, underscoring the role of GNN-LRP in dissecting the importance of higher-order interactions for the NNPs.

## NTL9
Finally, we show the GNN-LRP interpretation of an NNP-CG protein model, specifically that of 39 residue NTL9 (PDB ID: 2HBA, residues 1-39). We use the PaiNN architecture introduced in the previous section to learn a CG model of NTL9 from the same atomistic reference data used in a previous study[20], following the procedure introduced by Husic et al.[18]. More details on the training procedure can be found in the Methods Section. In this study, only $C_\alpha$ atoms are kept in the CG resolution and each amino-acid type is represented by a unique bead embedding. We selected this protein because of its well-characterized

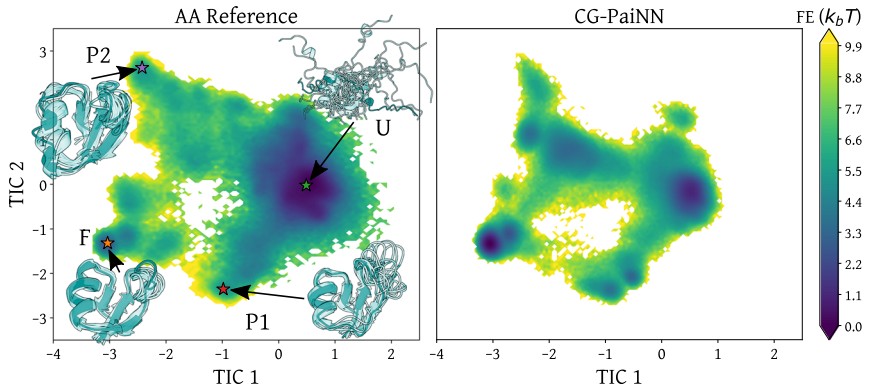

**Fig. 4 | Comparison of the free energy (FE) surface from the all-atom (AA, left) to the coarse-grained (CG, right) simulations shown as a function of the first two components of time-lagged independent component analysis (TICA)[69].** Four regions of interest in TICA space are labeled by F (folded), U (unfolded), P1

(folding pathway 1), and P2 (folding pathway 2). The structures used for the interpretation of the respective states are shown on the left. Protein visualizations generated with UCSF ChimeraX[109].

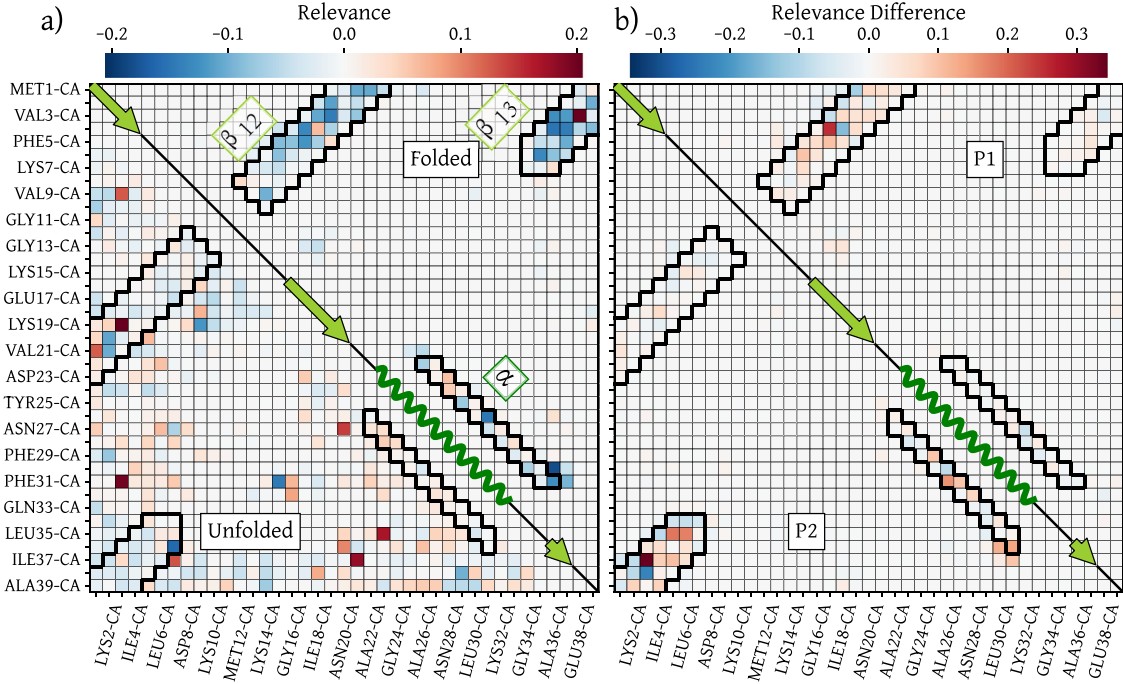

**Fig. 5 | Relevance contact maps for wild-type NTL9. a** Mean relevance of amino-acid pairs in the folded state (upper right) and the unfolded state (lower left); **b** Mean relevance difference between the folded state and the P1 (upper right) and P2 (lower left) states, respectively, i.e. $R_{P1/P2} - R_F$. The regions bordered in black

correspond to the contacts associated with the main secondary structure elements. The arrows and waves correspond to regions of the protein sequence inside a $\beta$-sheet and an $\alpha$-helix respectively.

folded/unfolded state and nontrivial folding pathways. A comparison of the Free Energy Surface (FES) projected onto the first two TICA components (collective variables capturing the slow motions of the system,[69]) is shown in Fig. 4.

The folded state of NTL9 contains 3 $\beta$-sheets that are formed by residues along the C- and N-terminal regions, as well as a central $\alpha$ helix (shown on the bottom left in Fig. 4). The stability of this short fragment of the N-terminal domain of the Ribosomal Protein L9 is likely due to the strong hydrophobic core between the $\beta$-sheets and the $\alpha$-helix[70,71]. To test whether the CG model is learning these effective interactions, we compute the relevance contribution for 2- and 3-body interactions in the trained model, for structures taken from different metastable states. The contact map in Fig. 5a shows the mean 2-body relevance contributions for each pair of amino-acids in the folded (upper right)

and unfolded (lower left) states. Interestingly, the 2-body interactions stabilizing the folded state correspond to contacts associated with the main secondary structure elements, while in the unfolded state both stabilizing and destabilizing interactions are found also outside of the secondary structure. The strongest 2- and 3-body interactions inside the folded state are shown in Fig. 6. The strongest 2-body contributions are found in the $\beta_{13}$ sheet, most of which are stabilizing interactions. Notably, the VAL3-GLU38 interaction is destabilizing, which indicates that the CG model learns to represent the side-chain specific interaction between the charged Glutamate and the hydrophobic Valine. This aligns with the Martini3 coarse-grained force-field parameterization for protein side-chains, where the interaction between the beads representing Valine and Glutamate side-chains is classified as "repulsive"[72,73]. The strongest 3-body interactions in the folded state,

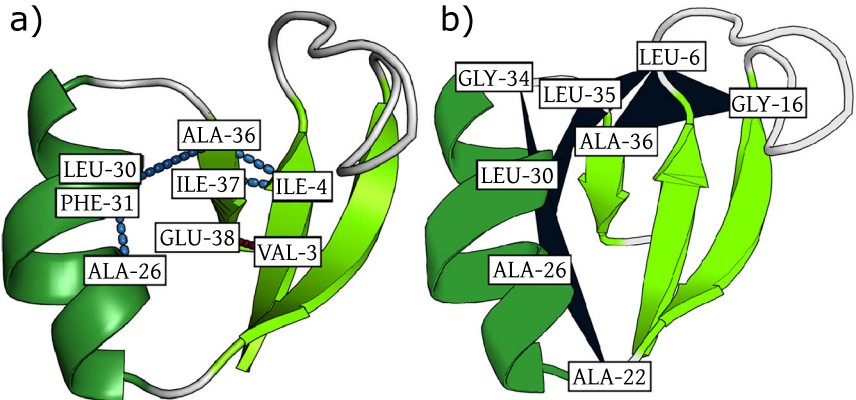

**Fig. 6 | Snapshots of folded NTL9 with the most relevant 2- and 3-body interactions learned by the coarse-grained model.** Panels **a** and **b** show the five most important 2- and 3-body interactions respectively. Blue lines indicate stabilizing interactions (negative relevance) and red lines destabilizing interactions (positive relevance). Darker color shades indicate stronger interaction strength (darker blue/red lines corresponds to stronger repulsive/attractive interactions). Visualizations generated with PyMOL[110].

shown in Fig. 6b, stabilize the helix and the overall tertiary structure of the protein. Note that because of the renormalization of the solvent degrees of freedom during coarse-graining, these interactions may not correspond to direct interactions between the residues but likely include some solvent-mediated interactions, such as hydrophobic effects. Indeed, the energy computed by the CG model corresponds to the CG potential of mean force, which is formally a free energy with both energetic and entropic contributions[74].

NTL9 can fold by two different pathways, which appear as two distinct "branches" in the free energy landscapes in Fig. 4. We examine the differences in relevance patterns between the two pathways connecting the folded to unfolded states. In panel b of Fig. 5 we show the difference in the mean relevance contributions of 2-body interactions in both intermediate states relative to the folded state. Here, a positive difference means that the interaction has a lower relevance in the folded state than in the intermediate state and thus that this interaction is less stable in the intermediate state. In the state indicated as P1 in Fig. 4, many interactions inside the $\beta_{12}$ sheet appear less relevant than in the folded state, whereas the difference with the folded state is essentially null in the $\beta_{13}$ sheet and in the $\alpha$-helix. This indicates that P1 corresponds to an intermediate state where the $\beta_{13}$ sheet and the $\alpha$-helix are native-like but the $\beta_{12}$ is less stable than in the native state. Indeed, analysis of the structures in P1 reveal that $\beta_{12}$ is not correctly formed. The P2 state shows the opposite behavior with interactions in the $\beta_{13}$ sheet and in the $\alpha$-helix less relevant than in the native state, indicating that in the P2 state, only the $\beta_{12}$ sheet is correctly formed. These two folding pathways with the same characteristics were also found in previous computational studies of NTL9[75–77]. Note that although the relevance attribution in the folded state shows that the network captures the interaction decay with the distance between residues, the relevance attribution in a given state provides much more information than contained in the contact maps for these states, as can be seen by comparing the relevance attribution of both intermediate states to their contact maps shown in Supplementary Fig. S11.

The interpretation of the learned interactions can be pushed a step further by considering the effects of mutations on the relevance analysis. We consider mutations of residues deemed stabilizing in the folded state. In the machine learned $C_\alpha$ CG model of the protein employed here, one can straightforwardly perform a mutation by changing the aminoacid identity, that is, by changing the embedding of the corresponding $C_\alpha$ bead. We select two mutations to illustrate the ability of the CG model to learn specific interactions such as hydrophobic/hydrophilic interactions between side-chains and side-chain specific packing. In particular, the mutation ILE4ASN is chosen to disrupt the hydrophobic interaction of the $\beta$-sheets, and the mutation LEU30PHE is chosen to disrupt the central $\alpha$-helix as well as the tight packing between the $\alpha$-helix and the $\beta$-sheets. These residues are flagged as important in the analysis above and have been shown in mutation experiments to play a role in the stabilization of the folded state[71,78].

In Fig. 7a we show the 2-body relevance contribution difference between the mutated states (LEU30PHE in the upper right and ILE4ASN in the lower left half) and the wild-type folded state. Replacing the identities of hydrophobic Isoleucine by polar Asparagine of about the same size at position 4 introduces a strong destabilization of all contacts with hydrophobic residues in neighboring sheets, as is also visualized in panel c. In the crystal structure of NTL9, LEU30 is tightly packed between the $\alpha$-helix and the $\beta$-sheets and mutation studies suggest that even a small change in side-chain size has a destabilizing effect[71]. Indeed, replacing the identity of Leucine 30 by the bigger Phenylalanine in our CG model induces a destabilization of the entire $\alpha$-helix (see Fig. 7a, b). Interestingly, the disruption also has an effect on contacts inside the $\beta_{13}$ sheet that do not directly involve the mutated residue, indicating that the model has indeed learned non-trivial many-body interactions. These findings further corroborate the ability of the CG model to learn amino-acid specific interactions in a $C_\alpha$-only representation.

## Discussion

In this work, we propose an extension of GNN-LRP to interpret the effective energy of machine learned CG models by dissecting it into interactions between sub-components of the systems. We have shown on the application to CG fluids that the learned interactions are physically meaningful and consistent even if different ML architectures are used. The explanations provided by GNN-LRP indicate when multi-body interactions are required to recover the thermodynamics of the fine-grained system, showing the suitability of this higher-order explanation method to CG NNPs. Moreover, the multi-body relevance contributions show that the different ML models have learned similar physically relevant interatomic interactions, indicating that these models effectively learn the same underlying potential energy surface. Additionally, GNN-LRP can uncover some small model artifacts invisible through plain MD simulation. We also showed that plain multi-body decomposition allows one to reach only the first one of these three conclusions, outlining the usefulness of GNN-LRP for explaining the NNP output. The application of this idea on an CG-NNP of a protein allows us to disentangle the strength of the different interactions between residues. It can also be used to evaluate the effect

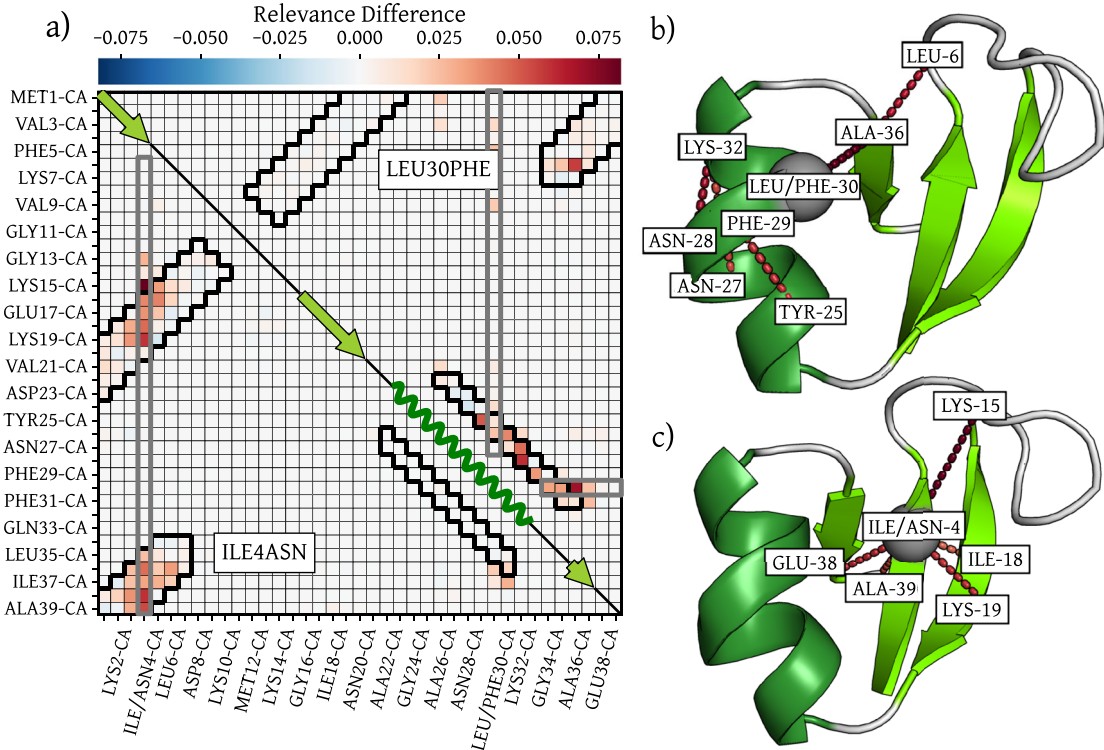

**Fig. 7 | Effect of mutations on the model prediction.** Panel **a** shows the mean 2-body relevance difference for each amino-acid pair with respect to the wild-type prediction, i.e. $R_{mut} - R_{WT}$. The upper right half corresponds to the LEU30PHE and the lower left to the ILE4ASN mutation. A red (positive) interaction corresponds to a higher relevance in the mutated state than in the wild-type, thus a destabilized interaction. The arrows and waves correspond to regions of the protein sequence inside a $\beta$-sheet and an $\alpha$-helix respectively. Panels **b**, **c** show an example structure with the mutated residue highlighted in gray as well as strongest interactions flagged by the network interpretation with colors corresponding to **a**. Visualizations generated with PyMOL[110].

of mutations in the model and shows that bottom-up $C_\alpha$-based CG NNPs capture amino-acid specific interactions without explicit representation of the side-chains.

These results provide reassurance that bottom-up CG NNPs indeed learn physically relevant terms by approximating the many-body potential of mean-force associated with the integration of part of the degrees of freedom[53]. We note that the methods introduced are model agnostic and can be used in general to interpret machine learned potentials of different systems at different resolutions.

This work suggests that CG NNPs can be trusted because of their ability to learn physically relevant terms, providing a strong argument for the wider use of these potentials despite their seemingly "black-box" nature. To this end, this work focused on previously well-studied systems, to act as validation of the method. The next natural step will be to apply these interpretation techniques to extract new scientific knowledge on previously unknown systems. Future work is still needed to refine these concepts to provide greater insight into ML models and allow researchers to be more systematic in their choice of architecture and functionalization. It is our hope that this work helps lay the groundwork for better understanding the outputs of NNP models as well as giving the coarse-graining community a way to probe these learned many-body effects more explicitly.

## Methods

### Coarse-Grained Neural Network Potentials
GNNs have been proposed as a promising method to learn interatomic potentials[10], and many different model architectures have been developed in recent years[79]. GNNs represent the underlying atomic details using structured graph data where nodes represent atoms and use the idea that locality dominates the energy landscape to draw

edges between nodes if two nodes are within a pre-defined cutoff distance. In GNNs there are generally three steps that go into producing the network output based on the input positions and node identities: (i) a message-passing step, where neighboring nodes (connected by edges) exchange information about their respective feature values, (ii) an update step, where the node features are modified based on the received messages, and (iii) a final readout step, where the features of each node are used to predict the target property[10]. Once the node features are fed through the readout layer, the network can make use of backpropagation to extract the force by taking derivatives of the energy based on molecule positions, which can then be used to train during force-matching or to propagate the dynamics in MD simulations.

In this manuscript, we examine two different GNN architectures for the effective CG energy: PaiNN[57] and SO3Net[58], two equivariant message-passing architectures that mainly differ on the order of their SO3-equivariant features ($l_{max} = 1$ for PaiNN and $l_{max} = 2$ for SO3Net). Both preserve the basic euclidean symmetries of the system, notably those of translation, rotation and reflection. Both architectures are parametrized to reproduce the CG potential of mean force using the force-matching approach[18,53,59–61]. Models were trained with SchNetPack[58,80] and simulations were done using the mlcg package[81]. As a comparison to more classical methods, IMC is also performed using the votca library[82] with results shown in Supplementary Section S3.

### Layer-wise Relevance Propagation
LRP has emerged as a method to explain model predictions in a post-hoc and model agnostic manner[34,50,83,84]. Originally, LRP has been used to obtain first-order explanations in the form of relevance attributions (also referred to as relevance scores) in the

input domain. E.g., for image classification tasks, the relevance attributions would indicate to what extent a respective pixel is responsible for the network decision[25,34]. The relevance attributions can be visualized in the input domain in form of a heat map, where large relevance attributions highlight features of the classified object that predominantly lead to the respective decision of the neural network[34]. Recently, efforts have been made to adapt LRP and other explanation methods to regression tasks[40,85,86], such as, e.g., the prediction of atomization energies[40,41].

Pixel-wise relevance attributions have contributed enormously to a better understanding of the inner workings of neural networks. However, in some cases, restricting explanations to first-order (input features) may result in oversimplified explanations. Especially for problems where the interaction between several input nodes is considerably strong, the relevance information of higher-order features becomes increasingly important. This is the case for the CG systems considered in this study, where multi-body interactions are essential[22–24,87]. A variety of higher-order explanation frameworks have been introduced[41,88–95]. One of those methods is GNN-LRP, which extends LRP to higher-order explanations for GNNs[41,90]. In the following, we will give a brief summary of the methodology of GNN-LRP, first describing first-order relevance propagation and then extending it to higher-order. For an in-depth introduction, please refer to[41,83].

We begin by following the path traced by Montavon et al. in[50,83]. We start with a single fully-connected layer of a neural network that connects $N_l$ to $N_{l+1}$ neurons. Each neuron in the $l+1$ layer is a multi-variable function $f : \mathbb{R}^{N_l} \to \mathbb{R}$, making it suitable for the relevance decomposition described in Equation (1). Our goal is to decompose the activity $\sigma(\mathbf{w}_\beta \mathbf{x} + b)$ of each neuron $\beta \in \{1, \ldots, N_{l+1}\}$ such that it resembles Eq. (1). We assume $\sigma(z) = \max(0, z)$, which is the ReLU function. In doing so we can stop the Taylor expansion at the first-order, obtaining, for a given neuron $\beta$:

$$\sigma(\mathbf{w}_\beta \mathbf{x} + b) = \sigma(\mathbf{w}_\beta \mathbf{x}^* + b) + \sum_{\alpha=1}^{N_l} \sigma'(\mathbf{w}_\beta \mathbf{x}^* + b)w_{\alpha\beta}(x_\alpha - x_\alpha^*) \quad (2)$$

Here, $w_{\alpha\beta}$ is the weighted connection from neuron $\alpha$ to $\beta$, and $x_\alpha$ and $x_\alpha^*$ are the components of the evaluation (or input) and expansion point, respectively. Assuming we start with the last layer in the neural network, we set $\sigma(\mathbf{w}_\beta \mathbf{x} + b) = R_\beta$, meaning relevance and activation are synonyms in this context. Our derivation, that is exact for a ReLU activation, is a first order approximation for other non linear functions such as SiLU, which is employed in the models throughout this study. Note however that since the decomposition is not exact anymore in this case, the sum of relevances is not equal to the total energy and we therefore do not attribute any energy unit to the relevance scores. Next, we need to fix the expansion point, $\mathbf{x}^*$. For now, the only constraint we require $\mathbf{x}^*$ to satisfy is $\mathbf{w}_\beta \mathbf{x}^* + b = 0$. In other words, $\mathbf{x}^*$ should belong to the so-called "ReLU hinge". This allows us to ignore the constant term in the Taylor expansion, thus simplifying it into an homogeneous linear transformation instead of an affine one, which makes all the calculations easier. Incorporating this assumption and computing the derivative leads to:

$$R_\beta(\mathbf{x}) = \sigma(\mathbf{w}_\beta \mathbf{x} + b) = \sum_{\alpha=1}^{N_l+1} \Theta[\mathbf{w}_\beta(\mathbf{x} - \mathbf{x}^*)]w_{\alpha\beta}(x_\alpha - x_\alpha^*) \quad (3)$$

Where $\Theta$ is the Heaviside step function that accounts for the evaluation on the proper side of the plane, and the sum now incorporates the bias term $b$ in the weight vector, mapping $\mathbf{x}^* \mapsto (\mathbf{x}^*, 1)$ and $\mathbf{w}_\beta \mapsto (\mathbf{w}_\beta, b)$. In this expression, each term represents the relevance

of input $\alpha$ in computing the activity value of neuron $\beta$, in formulae:

$$R_\beta(\mathbf{x}) = \sigma(\mathbf{w}_\beta \mathbf{x}) = \sum_{\alpha=1}^{N_l+1} \Theta[\mathbf{w}_\beta(\mathbf{x} - \mathbf{x}^*)]w_{\alpha\beta}(x_\alpha - x_\alpha^*) = \sum_\alpha R_{\alpha \leftarrow \beta}(\mathbf{x}) \quad (4)$$

If the relevance of input variable $\alpha$ is to be assessed, its contribution to each neuron in the following layer needs to be taken into account and summed together:

$$R_\alpha(\mathbf{x}) = \sum_{\beta=1}^{N_{l+1}} R_{\alpha \leftarrow \beta}(\mathbf{x}) \quad (5)$$

As mentioned earlier and in[83], different values of $\mathbf{x}^*$ will lead to different $R_{\alpha \leftarrow \beta}(\mathbf{x})$. As a result, the expansion point, that belongs to the ReLU hinge, can also explicitly depend on the input point, namely $\mathbf{x}^*(\mathbf{x})$. Interestingly, this allows the functional form of $R_{\alpha \leftarrow \beta}(\mathbf{x})$ to be further specified, revealing the explicit dependence on the activation (or relevance) of neuron $\beta$ (See Supplementary Section S6 for further details).

We can obtain an iterative rule for the relevance propagation from the last to the input layer of the neural network:

$$R_\alpha^l = \sum_\beta \frac{q_{\alpha\beta}}{\sum_\alpha q_{\alpha\beta}} \cdot R_\beta^{l+1}, \quad (6)$$

where the dependence on $\mathbf{x}$ is omitted and $q_{\alpha\beta}$ quantifies the contribution of neuron $\alpha$ to the activation of neuron $\beta$, incorporating the effect of the choice of the expansion point $\mathbf{x}^*(\mathbf{x})$. In this work, we choose the expansion point by using the so called "generalized $\gamma$-rule"[83,96] implying a certain value for $q_{\alpha\beta}$. It is an extension of the $\gamma$-rule originally introduced by Montavon et al.[83], to favor positive contributions in deep neural networks with rectifier (ReLU) nonlinearities. Details on such rule can be found in Supplementary Section S6, as well as a sketch of its derivations.

Since we are working with GNN architectures, the LRP approach must be adapted accordingly using GNN-LRP. The idea is to extend the LRP rule from a single layer transfer to the message-passing mechanism of a GNN. A message-passing step is composed of an aggregation and update step. The aggregation involves collecting information from neighboring nodes by summing or averaging their features, while the update mechanism integrates this aggregated information to update the node's features. The interaction between those two operations implies that, over different message-passing steps, information percolates between graph nodes by progressively mixing different components of the embedding vectors. The respective components of the embedding vectors are again named "neurons" in this context for consistency, but might be better known as "atomic features" to some readers from the field of NNPs. Consequently, instead of tracing the contribution between different layers, we should now trace the contribution coming from different components (neurons) of adjacent graph nodes weighted according to the related edge.

In contrast to the first-order node-wise relevance attributions, GNN-LRP explains the network prediction by assigning a relevance score to so-called "walks" inside the input graph. Each walk is a collection of connected nodes of the input graph, that we shall label with Latin indices. The length of the walks is dictated by the number of interaction layers (message passing) in the model. E.g., a model with two interaction layers would allow for up to 3-body walks. The concept of GNN-LRP is illustrated in Fig. 1a for a model with two interaction layers applied to an exemplary system composed of four particles (e.g., CG beads). The top part of the figure shows the forward pass of a

common GNN starting from the feature embedding on the input graph until the model output, while the bottom part of the figure illustrates how the walks with their associated relevance attributions are constructed. Note that the connections for message-passing in the GNN are defined by the graph structure, which is itself defined by the distance cutoff used.

Let us now describe the propagation procedure of GNN-LRP[41]. To conceptually extend the rule presented in Equation (6), consider a neuron that is part of the embedding vector, labeled with Greek indices within a fixed graph node. The relevance must be propagated through multiple nodes, similar to how it is done with layers in a fully connected network. At variance with this case, however, the layer step is replaced by the walk step over a given path in the main graph, which will be labeled by the tuple $(i, j, k, \ldots)$. For instance, $(i, j, k)$ represents the walk that starts from graph node $k$ and goes to node $i$ via node $j$.

Crucially the inter-node communication happens with two conceptually different steps: One - the aggregation step - that leverages the graph structure (employing link related information) but does not mix different Greek indices (neurons), and another one - the update step - that mixes different neurons on each node. Regarding the aggregation step, if $\lambda_\alpha^{ij}(\mathbf{r}^i - \mathbf{r}^j)$ is the edge-related feature that depends on the relative vector $\mathbf{r}^i - \mathbf{r}^j$ between the graph nodes $i$ and $j$, the following propagation rule is used:

$$R_\alpha^{ij} = \frac{\lambda_\alpha^{ij} x_\alpha^i}{\sum_i \lambda_\alpha^{ij} x_\alpha^i} R_\alpha^j. \tag{7}$$

Here, an entry in the node embedding $\mathbf{x}^i$ is denoted by the neuron $x_\alpha^i$. The equation above redistributes the relevance from one node to all the neighboring ones. Note that $\lambda_\alpha^{ij}$ is zero for node-pairs outside the cutoff radius, and unlike in Equation (6), we do not sum over output nodes $j$. For the neuron-mixing update step we can apply the propagation rule of Equation (6), keeping in mind that each of the values $q_{\alpha\beta}$ will now depend on the two-node interaction, namely $q_{\alpha\beta}^{ij}$. This happens as every graph node and every related neuron (namely the components of the vector embeddings) will have their own embedding value and therefore their own expansion point. This results in the following propagation rule for each message-passing step:

$$R_\alpha^{ij} = \sum_\beta \frac{\lambda_\alpha^{ij} q_{\alpha\beta}^{ij}}{\sum_i \sum_\alpha \lambda_\alpha^{ij} q_{\alpha\beta}^{ij}} \cdot R_\beta^j \tag{8}$$

With the above rule, relevance can be propagated through different message-passing steps and eventually be attributed to walks on the input graph. By progressively percolating the relevance of a given neuron in a given node $R_\alpha^k$ through the graph, one can construct the relevance of the walk $R_\alpha^{jk}$, then $R_\alpha^{ijk}$ and so on, until the maximum walk length, defined by the number of interaction blocks, is reached. Note that Equation (8) yields the walk relevance for a specific neuron $\alpha$. The total relevance of the respective walk is given by summing over all corresponding neurons:

$$R^{ijk} = \sum_\alpha R_\alpha^{ijk}. \tag{9}$$

In Supplementary Section S6, it is explained how the relevance attributions can be computed using efficient backpropagation algorithms. As depicted in Fig. 1, the $n$-body relevance can be obtained as a function of all the paths involving the sought nodes. More specifically in this paper, we post-process the relevance of each individual walk and aggregate it into contributions of subgraphs as explained in the following section.

## Postprocessing the Relevance

After computing the relevance for each individual walk inside the graph, we post-process the relevance scores to compute contributions of pairs and triplets of beads to the total energy, as illustrated in Fig. 1b. Indeed, since the relevance scores of individual walks are, under the named assumptions, terms in a sum decomposition of the network energy output, one can compute contributions to the total energy by summing over relevance scores of different walks.

To get the 2-body contribution of a pair of distinct beads $(i, j)$ to the total energy, named $\mathcal{R}_{ij}$, we simply sum over the relevances of the walks that sample only these two nodes, i.e.

$$\mathcal{R}_{ij} = \sum_{\mathcal{W} \in \{i,j\}^L, \, i \neq j} R^{\mathcal{W}} \tag{10}$$

where $L$ is the length of a walk (i.e. the number of interaction blocks plus one). $\{i, j\}^L$ denotes all the walks of length $L$ sampling nodes $i$ and $j$ only, i.e. $\{(a_1, \ldots, a_L) | a_m \in \{i, j\} \ \forall \ m \in \{1, \ldots, L\}\}$.

Similarly, to compute the 3-body contribution for a triplet of beads $(i, j, k)$ to the total energy $\mathcal{R}_{ijk}$, we compute the sum of the relevances of walks that sample only these three beads:

$$\mathcal{R}_{ijk} = \sum_{\mathcal{W} \in \{i,j,k\}^L, \, i \neq j \neq k} R^{\mathcal{W}} \tag{11}$$

where $\{i, j, k\}^L = \{(a_1, \ldots, a_L) | a_m \in \{i, j, k\} \ \forall \ m \in \{1, \ldots, L\}\}$.

## Simulation Details

**All-atom Simulations.** As reference data for the water and methane models, we use atomistic (AA) simulations with a periodic methane or TIP3P water box containing 125/258 atoms respectively. Only the heavy atoms were retained under the CG coordinate mapping: C for methane and O for water. All simulations were run in OpenMM[97] with Langevin dynamics at a temperature of 300K controlled with a damping coefficient of 1 ps⁻¹. Each system was simulated for 50 ns. For water, a 1 fs timestep was used with both coordinates and forces saved every 1 ps. These coordinates and forces are what are used in training the neural network model using force-matching. For methane, a timestep of 2 fs was employed and MD snapshots were saved every 2 ps. The water box was initiated using the openmmtools[98] testsystem WaterBox with parameter box_edge = 2 nm. The methane box was created using topotools[99] and simulated using the parameters in Supplementary Table S1.

The data used for the fast-folding variant of NTL9 (PDB ID 2HBA) is part of the dataset used in the previous study by Majewski et al.[20]. For convenience, the relevant details are briefly summed up here. NTL9 was solvated and ionized in a cubic box of side length 50 Å as in Ref. 77. ACEMD[100] and GPUGRID[101] were used to run MD simulations of the system using the CHARMM22star force-field[102] and the TIP3P water model[103] at 350 K. For production runs, a Langevin integrator with a timestep of 4 fs and a friction damping constant of 0.1 ps⁻¹ was used. Hydrogen to heavy atom bonds were holonomically constrained with 4 times hydrogen masses[104]. An MSM-based adaptive sampling approach[105] was used to enhance the sampling efficiency. From a total of 256 $\mu$s aggregated simulation time, about 2.4 million frames were used for training and validation.

**CG Simulations.** CG simulations were performed in the same way as Husic et al.[18], using Langevin dynamics with the BAOA(F)B integration scheme. The specific parameters for each system are summed up in Supplementary Table S2. Multiple independent simulations were run on a single GPU for efficient sampling.

## Neural Network Training

Coarse-grained models were trained using the same CGSchNet approach introduced in previous studies[18,61] but replacing the SchNet representation by equivariant models (PaiNN and SO3Net). In the CGSchNet approach, a thermodynamically consistent CG potential of mean force is learned via a reformulation of the force-matching approach[53,59,60] as a machine learning problem. The mapping operator for the CG forces called "basic aggregated" in[106] was chosen to be consistent with the presence of hydrogen-bond constraints in the training dataset. All training was done using SchNetPack 2.0[58].

For NTL9, a delta-learning approach is used where the CG energy is decomposed into $U(\mathbf{R}; \boldsymbol{\theta}) = U_{prior}(\mathbf{R}) + U_{net}(\mathbf{R}; \boldsymbol{\theta})$, with $U_{prior}$ based on physical intuition and where only $U_{net}$ is learned during training. The goal is to prevent bad model extrapolation in unphysical regions of the configuration space, that inherently lack training data.

For this protein, the prior has the form $U_{prior}(\mathbf{R}) = \sum_{bonds} U_{bond}(r) + \sum_{angles} U_{angle}(\theta) + \sum_{dihedrals} U_{dihedral}(\phi, \psi) + \sum_{non-bonded} U_{rep}(r)$. Here, $U_{bond}$, $U_{angle}$ and $U_{dihedral}$ are the same as used in previous studies[106], but the repulsive prior is set to $U_{rep}(r) = k \times \text{ReLU}((\sigma-r)^3)$ with $k = 20$ kcal/mol and $\sigma$ set to the minimum of the corresponding distribution in the training set, in order for its energy to be zero in the interpreted folded structures.

All models were trained using the AdamW optimizer[107] with a learning rate of $5.10^{-4}$ and a weight decay coefficient of 0.01. The hyperparameters for the different models are given in Supplementary Table S3.

## Data availability

Simulation data and scripts to reproduce the analysis and the plots shown in the manuscript are accessible on zenodo under https://zenodo.org/records/17068397. Source data are provided with this paper and can be found at https://box.fu-berlin.de/s/q4beCkaRHrY8Ac7. Source data are provided with this paper.

## Code availability

The codebase is available under MIT license at https://github.com/jnsLs/gnn-lrp-cg.git[108].

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

## Acknowledgements

We gratefully acknowledge funding from the Deutsche Forschungsgemeinschaft DFG (SFB/TRR 186, Project A12; SFB 1114, Projects B03, B08, and A04; SFB 1078, Project C7), the National Science Foundation (PHY-2019745), the Einstein Foundation Berlin (Project 0420815101), the German Ministry for Education and Research (BMBF) project FAIME 01IS24076, and the computing time provided on the supercomputer Lise at NHR@ZIB as part of the NHR infrastructure. K.R.M. was in part supported by the BMBF under grants 01IS14013A-E, 01GQ1115, 01GQ0850, 01IS18025A, 031L0207D, and 01IS18037A, and by the Institute of Information & Communications Technology Planning & Evaluation (IITP) grants funded by the Korean government (MSIT) No. 2019-0-00079, Artificial Intelligence Graduate School Program, Korea University and No. 2022-0-00984, Development of Artificial Intelligence Technology for Personalized Plug-and-Play Explanation and Verification of Explanation.

## Author contributions

C.C., K.R.M. and C.T. conceived the project, K.B., J.L. and C.T. trained, simulated and interpreted the models, D.R. conducted reference simulations, L.G. secured theoretical foundations, J.L., K.B. and C.T. wrote the code, K.B., J.L., C.T., L.G. and C.C. wrote the manuscript.

## Funding

## Competing interests

The authors declare no competing interests.
