## [Transparent Peer Review file · Nature Communications]

Peering inside the black box by learning the relevance of many-body functions in Neural Network potentials

Corresponding Author: Professor Cecilia Clementi

Version 0:

Reviewer comments:

Reviewer #1

(Remarks to the Author)

Bonneau et al. propose to better understand the output of a machine-learned (ML) coarse-grained (CG) potential. It makes use of a methodology initially proposed to analyze image classifiers, relying on contributions of any input to the final output throughout the neural network. The method is applied on several test systems: two homogeneous liquids and a protein. I find the increase in the Taylor expansion, which links to the n-body walks on the graph illustrated in Fig 1, meaningful and appealing. Unfortunately, the results fall short in novelty: most of the findings confirm what can be found in old literature, or what could likely be replicated with simple methodologies (more on that below). The Discussion section repeats conclusions that are well known in the CG literature. Granted, the emphasis here is on ML potentials, but there are no indications that the conclusions about CG-potential parametrization is any special with ML models. Perhaps the difficulty to find novel insight is not so surprising: unlike image classification where we have no formal theories, molecules abide to physics. The decomposition is clear: locality and body order---an aspect that the community at large, including this work, makes ample use of.

- The "relevance score" is never precisely defined. The text fuzzily describes it as a "contribution of their interaction to the total energy" or "essentially to a decomposition of the output energy." First, the term "total energy" suggests that the CG model outputs both a kinetic and a potential term, which I doubt it does. More broadly, is the score an n-body potential energy? Is it an n-body PMF? If it's an energy, why not define it formally and assign it units?
- The main conclusion about the homogeneous liquids seems to be that different ML models that reproduce the same pair and 3-body structure lead to similar decompositions. Previous CG models (with and without ML) on these systems have shown that the 2- and 3-body potentials look similar to what the relevance score outputs. This makes it a good validation, but does not yield new insight.
- The conclusions about the protein are somewhat different: they focus largely on pairwise contact maps. There, clear pairs are found to stabilize or destabilize select protein basins. I do wonder, though, how Fig 5 is really different from deriving a simple statistical potential: extracting a potential from the distribution of distances in the ensemble.
- The apparent stabilization of the secondary-structure elements is likely the result of the implicit-solvent modeling approach: entropic interactions are converted to enthalpic stabilization. This does not invalidate the results, but the lack of discussion is disappointing, given that the paper is about interpreting the CG model.
- p.12: "Notably, the VAL3-GLU38 interaction is destabilizing, which indicates that the CG model learns the side-chain specific interaction between the charged Glutamate and the hydrophobic Valine." There are two issues with this sentence. First, the main conclusion is trivial: we can all agree that a CG model capable of qualitatively describing the free-energy surface of a protein learns residue-specific interactions---it is not a mere homopolymer. Second, the analysis points at a *destabilizing* VAL-GLU interaction. What type of physical driving force involving a hydrophobic residue with a charged one are the authors referring to?

(Remarks on code availability)

I did not try to run the code. There is a README, and an example to run the code.

Reviewer #2

(Remarks to the Author)

The paper applies Layerwise-relevance propagation to coarse-grained neural network potentials in order to assess the presence and behaviour of 2-body and 3-body energy contributions to the total energy and to gain some insights into the dynamical behaviour of three systems: methane, water, and the protein NTL9.

Overall, the paper is well written and clearly presented, but the results it showcases are narrow in scope and, I believe, not particularly useful.

Its relevance appears to be restricted to specialized readers interested in the theory of neural network potentials, and I would recommend publication in a more specialized journal, even if the authors address my comments below.

I would like to add one caveat: since I am not an expert in protein simulation, I could have missed the relevance of some the analysis carried on in this manuscript on the protein NTL9. For this reason, if other reviewers find the paper publishable in this journal because of the protein analysis, I would suggest deferring to their opinion.

Below are two points I believe should be addressed to improve the quality of the manuscript.

a) It's not clear to me what the main advantage of the proposed method is over the widely employed analysis of the pair and triplet energy of isolated atoms/beads. It would be interesting to see if the same observations the authors draw on the water and methane systems could be gained by simply computing the energy of pairs and triplets of beads in different configuration. I would guess that, while 2-body relevance and direct energy computation differ mathematically, they would behave similarly as they both express how the system reacts to pairs of beads having a certain distance. This would also apply to triplets and angles, etc.

Could the authors comment on this? I would also suggest the authors to include such analysis in the supplementary information of the revised manuscript.

b) In page 7 of the main manuscript, the authors state "Note that the SO3Net model 111 exhibits irreducible representations up to an angular momentum of $l_{\max} = 2$, while 112 PaiNN utilizes a maximum angular momentum of $l_{\max} = 1$. As a consequence, in 113 comparison to SO3Net, PaiNN requires a larger cutoff to accurately reproduce the 114 RDF of water."

It is not obvious to me why the last sentence should be trivially true. Could the authors comment on this and provide an explanation in the main text or supplementary information?

(Remarks on code availability)

I have looked at the code and read the README, and praise the authors for providing detailed instructions and expected times to run. I have not run the scripts given that some parts require ~5 hours to run on a medium GPU.

Version 1:

Reviewer comments:

Reviewer #1

(Remarks to the Author)

The revision makes an argument that new insight was not the scope of the project, and was instead to show that neural networks learn physically relevant interactions. This argument is made multiple times throughout the reply. Unfortunately it is never substantiated or even defined precisely. The reply reads "ML-CG potentials should learn the same physics ... is not at all guaranteed"--of what type of other physics are we talking about? The only hint that the authors give is in deficiencies of image classification, where decisions were made on artefacts of the image. None of such surprises were found here.

Instead, the results confirm both old results of spline fitting from the structure-based CG community, as well as ML potentials at other resolutions. There are no "other physics" to be found--fitting of CG potentials with various methodologies and body orders has been done for a long time.

(Remarks on code availability)

Reviewer #2

(Remarks to the Author)

The authors significantly improved the manuscript, adding requested comparisons, widening the breadth of points discussed and delving more in depth into insights that can be drawn from the method they proposed and other existing force field analysis methods such as n-body analysis.

Despite the improved style, information content, and clarity, the core content of the manuscript remains unchanged.

Therefore, I still personally believe this article to be relevant only for a restricted audience of reader, and therefore perhaps more suited to a more specialized journal.

Nonetheless, machine learning force fields are becoming increasingly popular and widely used, to the point that even a specialized article such as this can be useful to many researchers.

I will, therefore, not give any recommendation regarding its relevance.

(Remarks on code availability)

Reviewer 1

Bonneau et al. propose to better understand the output of a machine-learned (ML) coarse-grained (CG) potential. It makes use of a methodology initially proposed to analyze image classifiers, relying on contributions of any input to the final output throughout the neural network. The method is applied on several test systems: two homogeneous liquids and a protein. I find the increase in the Taylor expansion, which links to the n-body walks on the graph illustrated in Fig 1, meaningful and appealing. Unfortunately, the results fall short in novelty: most of the findings confirm what can be found in old literature, or what could likely be replicated with simple methodologies (more on that below).

Author reply: *We are happy that the reviewer appreciates our approach of applying GNN-LRP to dissect learned n-body interactions. However, our main aim of applying explanation methods to coarse-grained (CG) neural network potentials (NNP) was not made clear in the previous version. It appears that the reviewers expected novel chemical insight into the considered systems. This was not the scope of this project. In contrast, we wanted to demonstrate the validity of this approach to provide insight into the n-body interactions learned by the NNP and show on the example of CG NNPs that neural network models learn physically relevant interactions. This is crucial since understanding the decisions of ML models increases trust in them and will in the future allow to detect malfunctioning in ML models, as well as extracting chemical insights when applied to unknown systems. We now emphasized our aim in our manuscript.*

We have added the following text:

- *in the abstract: “With these tools, neural network potentials can be practically decomposed into n-body interactions, allowing for interpretation similar to classical many-body potentials without compromising predictive power. We demonstrate the approach on three different coarse-grained systems including two fluids (methane and water) and the protein NTL9. The obtained interpretations suggest that well-trained neural networks learn physical interactions, which are consistent with theoretical principles.”*
- *on p.4: “we use Layer-wise Relevance Propagation applied to GNNs (GNN-LRP) to interpret the CG models beyond a mere energy prediction. In particular, we study the n-body contributions associated with the learned effective interactions in the NNP, and evaluate them based on fundamental principles”*
- *on p.5: “Here, we focus on the first step - enhancing trust in NNPs for CG systems - demonstrated in two examples.”*
- *on p.5: “The fact that the learned interactions align with existing physical and chemical knowledge makes the employed GNNs more trustworthy, supporting a wider use of these methods.”*

Also we address the detailed concerns of the reviewers (see below).

The Discussion section repeats conclusions that are well known in the CG literature. Granted, the emphasis here is on ML potentials, but there are no indications that the conclusions about CG-potential parametrization is any special with ML models. Perhaps the difficulty to find novel insight is not so surprising: unlike image classification where we have no formal theories, molecules abide to physics. The decomposition is clear: locality and body order—an aspect that the community at large, including this work, makes ample use of.

Author reply: *We thank the reviewer for this comment, and while we agree on the fact that ML-CG potentials should learn the same physics that is used to design classical CG potentials, we would like to underline that this is not at all guaranteed, nor has been demonstrated before, and it is not widely accepted by the community. On the contrary, a large fraction of the community still considers ML potentials a black box and is concerned that the model may just memorize the data and not be suitable for extrapolation and applications to unknown systems. Our manuscript clearly shows, for the first time, that properly designed CG NNPs learn the same physical principles used in the design of classical potentials. To the best of our knowledge, this has never been shown before, and it is the real novelty of this work.*

We believe that demonstrating that the ML models learn physically meaningful interactions that align with the heuristics implemented in parameterized force fields is a significant result. The fact that the ML model actually learns the physics that the molecules abide to shouldn't be taken for granted: In principle, the large number of degrees of freedom and parameters employed in the neural network would also allow for learning "unphysical" interactions, that would yield similar results on validation sets but might lead to model failure when making predictions on different conformations. With the approach we propose, we can investigate the learned interactions beyond an educated guess. This information is typically hidden in the ML model and not accessible by standard approaches such as the many-body decomposition framework, as we now explicitly show (see below).

- The "relevance score" is never precisely defined. The text fuzzily describes it as a "contribution of their interaction to the total energy" or "essentially to a decomposition of the output energy."

Author reply: *We agree with the reviewer that in the previous version of the manuscript the relevance score was not clearly described. We now added both an intuitive explanation, and a complete mathematical derivation of GNN-LRP in the Methods section of the main text (Section 4.2) and in the Supplementary Section S6, respectively. We describe how to derive GNN-LRP starting from a Taylor expansion and give an intuition for the different propagation rules of LRP while motivating our choices. In addition we have added Section 4.3 in the main text, which describes the postprocessing of the obtained relevance attributions (scores). This, in combination with Fig. 1 should provide a clear definition of the relevance score and in particular of the n-body relevance contributions analyzed for the three examples water, methane and NTL9.*

First, the term "total energy" suggests that the CG model outputs both a kinetic and a potential term, which I doubt it does.

Author reply: *We agree with the reviewer, and to avoid ambiguity we now use the term "potential energy".*

More broadly, is the score an n-body potential energy? Is it an n-body PMF? If it's an energy, why not define it formally and assign it units?

Author reply: *GNN-LRP in combination with our postprocessing gives insight into n-body interactions learned by the NNP. However, the relevance of the n-body interactions is not equivalent to the energy of isolated n-bodies such as in a many-body expansion. The relevance score is a measure of how much a substructure of the graph contributes to the potential energy given its chemical environment (taking into account the surrounding particles). Hence, the n-body relevance is not equivalent to the n-body potential energy. We discuss this point explicitly in the main text, in several places:*

- on p.7: *“In a nutshell, GNN-LRP decomposes the model’s energy output into relevance attributions to sequences of graph edges. Those sequences describe “walks” over a few nodes in the input graph. The associated relevance attribution is also often referred to as the relevance score of the walk. By aggregating the relevance scores of all walks associated with a particular subgraph, we can determine its n-body relevance to the model output [...]”*

and in the Supplementary Section S4:

- *“[...] the “n-body contributions” computed by GNN-LRP are not the same as the n-body energy terms of the many-body expansion. Indeed, GNN-LRP is decomposing the total energy of a full frame into contributions for different walks inside the graph, taking also into account the surroundings of each subset of nodes included in a walk. In contrast, the many-body decomposition computes the n-body energy from isolated beads/molecules, thus ignoring the effect of the neighboring molecules in a bulk environment.”*

We emphasized the difference between our GNN-LRP approach and the commonly used many-body expansion framework in the main text:

- pp.4-5: *“A key advantage of using GNN-LRP for this task is its ability to reveal learned interactions among a subset of beads taking into account their surroundings. On the other hand, the many-body decomposition of the model’s output with the traditional recursive method [...] determines the energy needed to form isolated n-mers of beads from sub-elements, thus ignoring the effect of the surrounding environment.”*
- p.14: *“As mentioned in the Introduction Section 1, the major advantage of GNN-LRP over traditional many-body decomposition is that GNN-LRP gives n-body contributions of subgraphs in the environment of the entire system while determining the learned n-body interactions by means of the many-body decomposition framework computes the n-body energies of isolated n-mers. The latter gives rise to a technical issue that becomes clear for the above fluids methane and water. In the many-body decomposition the system’s energy is expressed in terms of n-body energies. Those n-body energies represent the energy to form isolated 1,2,...,n-mers from m-mers with $m < n$. In order to provide a many-body decomposition of the learned effective potential energy, the energies of individual n-mers need to be predicted by the NNP. However, the training data of the NNP does not include isolated n-mers, it only includes bulk frames, which is why the resulting many-body decomposition has a high chance of being noisy and does not provide physical insight.”*

Additionally, in the Supplementary Section S4, we explicitly demonstrate that a plain many-body expansion in terms of isolated n-bodies does not provide a robust interpretation of the trained NNP. We do not use energy units because the relevance score is only approximately a decomposition of the energy, its calculations involve some approximations (discussed in the Methods section and Supplementary Section S6) and the total relevance does not exactly match the total energy. This is now discussed in the Methods section.

- The main conclusion about the homogeneous liquids seems to be that different ML models that reproduce the same pair and 3-body structure lead to similar decompositions. Previous CG models (with and without ML) on these systems have shown that the 2- and 3-body

potentials look similar to what the relevance score outputs. This makes it a good validation, but does not yield new insight.

Author reply: *We thank the reviewer for noting that the GNN-LRP analysis of the bulk liquids is a good validation of the explanation framework. As mentioned in the replies to the comments above, the main aim of this work was not clearly explained in the previous version of the manuscript. We (and the community in general) do not take for granted that NNPs for molecules learn the physics that these molecules abide to. A lot of work has been done, for example on image classification, to show that some neural networks may yield seemingly great performance due to an undetected bias in the training set, leading to very poor performance in new prediction tasks. In order to enable trusting the more widely used NNPs, we here aim to demonstrate the capability of machine-learned coarse-grained models to learn physically relevant interactions, independently from their performance in MD simulation. To this end we chose previously well-studied systems on purpose, to validate our explanation method, compare the learned n -body interactions of different ML models and discuss them in the context of known theory. We consider an exciting result that our ML models learn similar features and the learned interactions align with theory and empirical evidence, which we emphasize is not a trivial conclusion. We hope that the scope of this work is clear after having adapted the main text.*

- The conclusions about the protein are somewhat different: they focus largely on pairwise contact maps. There, clear pairs are found to stabilize or destabilize select protein basins. I do wonder, though, how Fig 5 is really different from deriving a simple statistical potential: extracting a potential from the distribution of distances in the ensemble.

Author reply: *We thank the reviewer to raise this point. We have now evaluated the distance distribution in different ensembles (shown in Supplementary Fig. S11) and compared with the 2-body relevance contribution of pairs of amino-acids assigned by the trained NNP to different conformations of protein NTL9. Interestingly, this comparison clearly shows that, especially on the intermediate states P1 and P2, the distribution of distances does not reflect the GNN-LRP interpretation, emphasizing that the model is not simply learning mere statistics on distances present in the training dataset.*

- The apparent stabilization of the secondary-structure elements is likely the result of the implicit-solvent modeling approach: entropic interactions are converted to enthalpic stabilization. This does not invalidate the results, but the lack of discussion is disappointing, given that the paper is about interpreting the CG model.

Author reply: *We agree with the reviewer that through the renormalization of degrees of freedom associated with coarse-graining (and thus the implicit solvent approach), the CG potential of mean force (PMF) used to propagate the dynamics in a thermodynamically consistent way is a formal free energy that has both energetic and entropic contributions and both terms are needed to effectively capture solvent-mediated interactions. We have added a short discussion in the main text to make this point clear:*

- *p. 16: Note that because of the renormalization of the solvent degrees of freedom during coarse-graining, these interactions may not correspond to direct interactions between the residues but likely include some solvent-mediated interactions, such as hydrophobic effects. Indeed, the energy computed by the CG model corresponds to the CG potential of mean force, which is formally a free energy with both energetic and entropic contributions [...].*

However, separating the entropic and energetic contributions requires an explicit temperature-dependence of the PMF, which is only achievable with additional training data at different temperatures and a different architecture, a non-trivial framework on which our group is currently working and that is beyond the scope of this work. For this reason, a more detailed analysis on the entropic or energetic nature of the stabilizing interactions is not included in this manuscript.

- p.12: "Notably, the VAL3-GLU38 interaction is destabilizing, which indicates that the CG model learns the side-chain specific interaction between the charged Glutamate and the hydrophobic Valine." There are two issues with this sentence. First, the main conclusion is trivial: we can all agree that a CG model capable of qualitatively describing the free-energy surface of a protein learns residue-specific interactions—it is not a mere homopolymer. Second, the analysis points at a *destabilizing* VAL-GLU interaction. What type of physical driving force involving a hydrophobic residue with a charged one are the authors referring to?

Author reply: *We agree with the reviewer that a mere homopolymer would not reproduce the free energy landscape of the protein, not even qualitatively. However, we do not think it is trivial that the learned representation of the interactions between the beads by the model can present distinct features of the side-chains related to hydrophobicity, acidity, or size, as it is suggested by the mutational analysis on the protein. The 2-body interactions learned by the CG model are not necessarily physical interactions, but are effective interactions including the effects of the renormalized degrees of freedom. Dissecting the entropic or energetic nature of the interactions requires an energy/entropy decomposition of the learned free energy, that is not possible with the current framework, as discussed above. Additionally, an effective destabilizing VAL-GLU interaction is in agreement differently parametrized coarse-grained models, such as the Martini3 model, where the interaction between beads representing both side-chains is deemed "repulsive" (see citations in the main text).*

Reviewer 2

The paper applies Layerwise-relevance propagation to coarse-grained neural network potentials in order to assess the presence and behaviour of 2-body and 3-body energy contributions to the total energy and to gain some insights into the dynamical behaviour of three systems: methane, water, and the protein NTL9.

Overall, the paper is well written and clearly presented, but the results it showcases are narrow in scope and, I believe, not particularly useful. Its relevance appears to be restricted to specialized readers interested in the theory of neural network potentials, and I would recommend publication in a more specialized journal, even if the authors address my comments below.

I would like to add one caveat: since I am not an expert in protein simulation, I could have missed the relevance of some the analysis carried on in this manuscript on the protein NTL9. For this reason, if other reviewers find the paper publishable in this journal because of the protein analysis, I would suggest deferring to their opinion.

Author reply: *We thank the reviewer for their constructive comments. We believe there might have been a misunderstanding on the scope of this manuscript which we hope to have made clearer in the present version. Our goal was not to provide new insight on already well-studied systems such as water, methane and the protein NTL9. Rather, the manuscript goal is to show that increasingly popular ML force-fields, especially for their use in CG force-fields,*

learn physically meaningful interactions. The systems considered in the manuscript were chosen precisely because they are well-studied, to have a clear ground truth to which to compare what the NNP learns. The novelty in this manuscript does not lie on the insight into the specific physical systems, but on the fact that NNPs learn physically relevant interactions.

We have now modified the manuscript to make our goal clear, especially in the introduction and discussion sections:

- Introduction, p. 4: “For trusting image classifiers, it is important to know if an accurate classification stems from a correct learning of the features or a learning of an undetected bias in the training set [...]. Analogously, for trusting NNPs and their ability to extrapolate to new systems, it is important to know if an accurate prediction arise from the network learning the physical properties of the different interactions or is merely a data memorization, or a compensation of errors [...].”
- Introduction, p. 5: “Ideally, an interpretable model should enable researchers to build trust in its predictions and, in a second step, extract scientific knowledge from successful applications while identifying the sources of deficiencies or anomalies when the model fails. Here, we focus on the first step - enhancing trust in NNPs for CG systems by demonstrating the physically sound nature of the learned interactions in two examples.”
- Introduction, p. 5: “The fact that the learned interactions align with existing physical and chemical knowledge makes the employed GNNs more trustworthy, supporting a wider use of these methods.”
- Discussion, p. 21: “This work supports that CG NNPs can be trusted because of their ability to learn physically relevant terms, providing a strong argument for the wider use of these potentials despite their seemingly “black-box” nature. To this end, this work focused on previously well-studied systems, to act as validation of the method. The next natural step will be to apply these interpretation techniques to extract new scientific knowledge on previously unknown systems.”

Below are two points I believe should be addressed to improve the quality of the manuscript.

a) It’s not clear to me what the main advantage of the proposed method is over the widely employed analysis of the pair and triplet energy of isolated atoms/beads. It would be interesting to see if the same observations the authors draw on the water and methane systems could be gained by simply computing the energy of pairs and triplets of beads in different configuration. I would guess that, while 2-body relevance and direct energy computation differ mathematically, they would behave similarly as they both express how the system reacts to pairs of beads having a certain distance. This would also apply to triplets and angles, etc. Could the authors comment on this? I would also suggest the authors to include such analysis in the supplementary information of the revised manuscript.

Author reply: We thank the reviewer for this valuable comment. We have followed the reviewer’s suggestion and performed a many-body decomposition of the learned NNPs for water and methane and we report the results in the Supplementary Information (Section S4). Very interestingly, the results are quite different from what is obtained with GNN-LRP. The NNPs do not incorporate analytical expressions of n-body interactions but they learn those

interactions implicitly based on the chemical environment of each particle. GNN-LRP allows for extracting a measure of n -body interactions given the particular chemical environment and it is conceptually different from the many-body decomposition into isolated n -body energies: GNN-LRP allows to extract the relevance of n -body interactions even though the model is trained on reproducing the thermodynamics of the entire system. The training data does not include isolated n -bodies with corresponding energy targets, and for this reason plain many-body decomposition is not very meaningful for NNPs.

We emphasized the difference between our GNN-LRP approach and the more standard many-body decomposition framework in the main text:

- pp. 4-5: “A key advantage of using GNN-LRP for this task is its ability to reveal learned interactions among a subset of beads taking into account their surroundings. On the other hand, the many-body decomposition of the model’s output with the traditional recursive method [...] determines the energy needed to form isolated n -mers of beads from subelements, thus ignoring the effect of the surrounding environment.”
- p. 14: “As mentioned above, the major advantage of GNN-LRP over traditional many-body decomposition is that GNN-LRP gives n -body contributions of subgraphs in the environment of the entire system while determining the learned n -body interactions by means of the many-body decomposition framework computes the n -body energies, as predicted by the NNP, to form isolated n -mers from m -mers with $m < n$. As demonstrated and discussed in detail in Supplementary Section S4, the traditional many-body decomposition of the NNP gives rise to noisy and uninformative 2- and 3-body terms as these terms are taken in isolation and not in the context of the bulk fluid, as is instead done by GNN-LRP.”

and in the Supplementary Section S4, where we explicitly show in the case of methane and water how a many-body decomposition in terms of isolated n -bodies does not provide a meaningful interpretation of the NNP.:

- “As one can see, the “ n -body contributions” computed by GNN-LRP are not the same as the n -body energy terms of the many-body decomposition. Indeed, GNN-LRP is decomposing the total energy of a full frame into contributions for different walks inside the graph, taking also into account the surroundings of each subset of nodes included in a walk. In contrast, the many-body decomposition computes the n -body energy from isolated beads/molecules, thus ignoring the effect of the neighboring molecules in a bulk environment.”

b) In page 7 of the main manuscript, the authors state "Note that the SO3Net model exhibits irreducible representations up to an angular momentum of $l_{\max} = 2$, while PaiNN utilizes a maximum angular momentum of $l_{\max} = 1$. As a consequence, in comparison to SO3Net, PaiNN requires a larger cutoff to accurately reproduce the RDF of water.". It is not obvious to me why the last sentence should be trivially true. Could the authors comment on this and provide an explanation in the main text or supplementary information?

Author reply: We thank the reviewer for pointing out that this is not a trivial connection. The irreducible representations of SO_3 allows for a more expressive network, that can parametrize more complex functions on the unit sphere. As a consequence, SO3Net can learn more complex chemical environments. With decreasing rotation order, PaiNN has a reduced

expressivity and requires to include a larger neighborhood to describe the chemical environment. We added a description in the main text

- *With increasing l_{\max} , the NNP can learn more expressive representations of the chemical environment in each message-passing block and as a consequence, in comparison to SO3Net, PaiNN requires a larger cutoff to accurately reproduce the RDF of water.*

and we provide a more detailed explanation in the Supplementary Section S3.1.

Reviewer 1

The revision makes an argument that new insight was not the scope of the project, and was instead to show that neural networks learn physically relevant interactions. This argument is made multiple times throughout the reply. Unfortunately it is never substantiated or even defined precisely. The reply reads "ML-CG potentials should learn the same physics ... is not at all guaranteed"—of what type of other physics are we talking about? The only hint that the authors give is in deficiencies of image classification, where decisions were made on artefacts of the image. None of such surprises were found here. Instead, the results confirm both old results of spline fitting from the structure-based CG community, as well as ML potentials at other resolutions. There are no "other physics" to be found—fitting of CG potentials with various methodologies and body orders has been done for a long time.

Author reply: *We acknowledge the reviewer’s standpoint and respectfully disagree with it. If underlying data was generated with a specific function it is not guaranteed that a fitting method can necessarily recover that function, especially when the fitting function is as expressive as a neural networks. Because molecules abide to physics it is not a given that fitted CG force-fields should recover the physics. This has been shown extensively in literature provided in the manuscript, not only in image classification but also for MLFFs, see for example Fig. 3 in [1].*

Reviewer 2

The authors significantly improved the manuscript, adding requested comparisons, widening the breadth of points discussed and delving more in depth into insights that can be drawn from the method they proposed and other existing force field analysis methods such as n-body analysis.

Despite the improved style, information content, and clarity, the core content of the manuscript remains unchanged. Therefore, I still personally believe this article to be relevant only for a restricted audience of reader, and therefore perhaps more suited to a more specialized journal.

Nonetheless, machine learning force fields are becoming increasingly popular and widely used, to the point that even a specialized article such as this can be useful to many researchers.

I will, therefore, not give any recommendation regarding its relevance.

Author reply: *We thank the reviewer for noting the improvement in the paper. As they point out, machine-learned force-fields (MLFFs) have gained remarkable popularity due to their unparalleled performance in predicting molecular properties. However, their increasing adoption has also raised concerns about their “black-box” nature, which can hinder trust in predictions and limit physical insights into the systems under study. We believe MLFFs are revolutionizing the field of molecular dynamics thanks to their ability to address long-standing challenges in accuracy and computational efficiency. In this context, we believe our work is particularly timely and relevant: it not only enhances confidence in MLFF predictions but also paves the way for extracting meaningful physical insights from these powerful models.*

References

- [1] Wang, J. *et al.* Machine learning of coarse-grained molecular dynamics force fields. *ACS Cent. Sci.* **5**, 755–767 (2019).